# Benchmarking Smoothness and Reducing High-Frequency Oscillations in Continuous Control Policies

## Abstract

Reinforcement learning (RL) policies are prone to high frequency oscillations, specially undesirable when deploying to hardware in the real-world. In this paper, we identify, categorize, and compare methods from the literature that aim to mitigate high frequency oscillations in RL. We define two broad classes: loss regularization and architectural methods. At their core, they incentivize learning a smooth mapping, such that nearby states in the input space produce nearby actions in the output space. We present benchmarks in terms of policy performance and smoothness with staple RL environments from Gymnasium, as well as two robotics locomotion tasks that include deployment and evaluations in the real-world. Finally, we also propose hybrid methods that combine elements from both loss regularization and architectural methods, and outperform the existing approaches in the simulation benchmarks as well as in the real-world.

## 1 Introduction

Reinforcement learning (RL) policies are prone to high frequency oscillations. When no limitations or constraints are imposed in either the learning or in the environment, RL agents easily develop exploitative behavior that maximizes reward to the detriment of everything else. While chasing high task performance (reward) is the goal of learning, there are scenarios where other factors must also be considered. For example, when deploying a policy to hardware in the real-world high-frequency oscillations are specially undesirable as they can cause damage.

A straightforward way to mitigate the issue is to include a penalization term as part of the reward term. However, the learning algorithm tends to exploit the reward function and it can lead to situations where the policy has subpar performance. For many tasks it is difficult to design a reward function in the first place. Adding a penalization term for high frequency oscillations essentially modifies the original learning objective, and can be difficult to tune. If the penalization weight is too large, the agent might prefer to not do much to avoid large negative rewards. On the other hand, if the weight is too small it might choose to ignore it and still generate high-frequency oscillations. Ideally, a method should allows us to keep the original learning objective and avoid adding new elements of complexity to the reward design.

Another approach to reduce high-frequency oscillations is to filter the actions outputted from the policy, for example with a low-pass filter. Thinking of the classic agent – environment diagram in RL (Sutton & Barto, 2018), we argue that this type of approach is effectively adding a constraint to the environment rather than the agent (or policy) itself. In fact, filtering the actions could lead to even greater oscillations in the raw outputs of the policy. That being said, filtering is effective and quite common in practice, specially in robotics applications (Peng et al., 2020). The biggest issue when using a traditional filter is that it has memory. This means that if the observation space does not include past actions and past observations the policy will not be able to learn an effective model, as it violates the MDP assumption (Van Otterlo & Wiering, 2012). Also discussed in Mysore et al. (2021), this leads to a larger model in terms of parameters and complexity.

In this work, we categorize, adapt, and compare methods that aim to mitigate the problem of high frequency oscillations in RL. We focus on methods that do not rely on reward penalization, or environment modifications such as post processing actions. Rather, we identify two classes of methods

in the literature: loss regularization and architectural methods. At their core, the methods incentivize or impose constraints such that the policy learns to produce smooth mappings between input and output space. A smooth mapping is usually expressed as having states that are nearby in input space produce nearby actions in output space (Shen et al., 2020). A common mechanism employed by multiple existing works is to constrain the upper bound of the Lipschitz constant of the policy network (Fazlyab et al., 2019), either globally Liu et al. (2022) or in a local manner (Kobayashi, 2022; Song et al., 2023).

We benchmark multiple algorithms on classic RL environments in terms of policy performance and action smoothness with Gymnasium (Towers et al., 2023). Additionally, we also benchmark on two robotics locomotion tasks, including evaluation and deployment of the best-performing method in the real-world. Finally, we propose new, hybrid methods that outperform the existing approaches. Our contributions can be summarized as follows:

- Benchmarking existing methods both in classic RL environments and more application focused environments;
- Categorizing existing methods in two broad classes: loss regularization methods and architectural methods;
- Proposing new hybrid methods that combine elements from other existing methods.

## 2 RELATED WORKS AND METHODS CATEGORIZATION

**Benchmarking in RL**. Reinforcement learning is a diverse field with a large variety of tasks and algorithms. With a large array of possibilities it is common for practitioners to look for benchmarks to aid in algorithm selection. Duan et al. (2016) presented a benchmark for continuous control policies in several classic tasks as well as more complex tasks such as humanoid locomotion in 3D simulation. Other works have performed benchmarks that focus in domains such as meta reinforcement learning (Yu et al., 2020), manipulation tasks (Fan et al., 2018), and real-world deployment (Mahmood et al., 2018; Gürtler et al., 2023). In the context of smooth policies, past works have presented a few benchmarks (Song et al., 2023; Kobayashi, 2022; Mysore et al., 2021), but this usually takes the form of direct comparisons with previous methods. In our work, we aim to fill the comparison gap with more extensive comparisons between methods that learn smooth policies. In the remainder of this section we introduce and describe two classes along with existing works that fit this description.

### 2.1 LOSS REGULARIZATION METHODS

Loss regularization methods aim to reduce the oscillation frequency of the actions by adding regularization components to the standard RL loss function. They take the form of

$$\mathcal{L} = \mathcal{L}_{\text{RL}} + \mathcal{L}_{Reg} \tag{1}$$

where $\mathcal{L}_{\text{RL}}$ is a policy gradient loss such as PPO (Schulman et al., 2017), TRPO (Schulman et al., 2015), and similar methods; and $\mathcal{L}_{Reg}$ is the regularization loss. We investigate two recent works that propose loss regularization methods.

**CAPS (Mysore et al., 2021)**. Uses two regularization components. The first is a temporal component $\mathcal{L}_T$, which minimizes the distance between the actions of two consecutive states $s_t$ and $s_{t+1}$. The second is a spatial component $\mathcal{L}_S$ that minimizes the difference between the state $s_t$ and a state $\bar{s}_t$ sampled from a normal distribution in the neighborhood of $s_t$. This takes the form of

$$\begin{aligned}
\mathcal{L}_T &= D(\pi_\theta(s_t), \pi_\theta(s_{t+1})) \\
\mathcal{L}_S &= D(\pi_\theta(s_t), \pi_\theta(\bar{s}_t)), \quad \text{where } \bar{s}_t \sim \mathcal{N}(s_t, \sigma) \\
\mathcal{L}_{\text{CAPS}} &= \lambda_T \mathcal{L}_T + \lambda_S \mathcal{L}_S
\end{aligned} \tag{2}$$

where $\pi_\theta$ is the actor network, $D(\cdot)$ is a distance function, and $\lambda_T$, $\lambda_S$, and $\sigma$ are hyperparameters to be tuned. This method is similar to the one proposed by Shen et al. (2020), with the main distinction

that CAPS uses the L2 distance between sampled actions, while Shen et al. (2020) employed KL divergence on the output distributions.

For the overlapping environments we use the same hyperparameters from the original paper (Mysore et al., 2021). For the new locomotion environments we performed a short hyperparameter search and chose the best values. The complete hyperaparameter details are presented in the Appendix Tables 2, 5, and 9.

**L2C2 (Kobayashi, 2022)**. Uses two regularization components with a similar mechanism to the spatial component in CAPS. Notably, the regularization is employed both to the outputs of the actor network $\pi_\theta$ and the value network $V_\theta$. Moreover, the sampling distance is bounded relative to the distance of two consecutive states $s_t$ and $s_{t+1}$, rather than a hyperparameter. The L2C2 regularization is computed in the following way

$$
\begin{aligned}
\bar{s}_t &= s_t + (s_{t+1} - s_t) \cdot u, \quad \text{where } u \sim \mathcal{U}(.) \\
\mathcal{L}_{s,\pi} &= D(\pi_\theta(s_t), \pi_\theta(\bar{s}_t)) \\
\mathcal{L}_{s,V} &= D(V_\theta(s_t), V_\theta(\bar{s}_t)) \\
\mathcal{L}_{\text{L2C2}} &= \lambda_\pi \mathcal{L}_{s,\pi} + \lambda_V \mathcal{L}_{s,V}
\end{aligned}
\tag{3}
$$

where $\mathcal{U}$ is a uniform distribution, $D$ is a distance metric, $\pi_\theta$ and $V_\theta$ are the actor and value network, and $\lambda_\pi$ and $\lambda_V$ are weights for each regularization component. For brevity, the uniform sampling details and its hyperparameters are omitted here. We invite the reader to read the original work from Kobayashi (2022) for an in-depth discussion of the state sampling and definition of the hyperparameters. Our training hyperparameter choices are presented in the Appendix Tables 2, 5, and 9.

**L2C2** and **CAPS** are similar, with the main difference coming from the sampling method. One could argue that the temporal element of **CAPS** is redundant, since a state that is sampled nearby and two consecutive states should produce more or less the same regularization signal. As such, **L2C2** drops the temporal element in favor of optimizing both the actor and the value network outputs.

## 2.2 ARCHITECTURAL METHODS

Architectural methods aim to reduce the oscillation frequency of the actions by modifying the learning components of the network. In the case of the Lipschitz based methods they also add an element to the loss function. However, the objective function is used to bring down the upper bound of the Lipschitz value of the network, rather than directly optimizing for action differences or the mapping function as in the loss regularization category.

**Spectral Normalization – Local SN (Takase et al., 2022)**. Spectral normalization is commonly used to stabilize the training of Generative Adversarial Networks (Miyato et al., 2018). It consists of a rescaling operation on the weights of a network layer by its spectral norm $\sigma(\boldsymbol{W})$. The normalized weights are simply given by $\boldsymbol{W}_{SN} = \delta \cdot \frac{\boldsymbol{W}}{\sigma(\boldsymbol{W})}$. In the context of reinforcement learning, Takase et al. (2022) proposed global and local variants of the spectral normalization. The difference between the global and local variants, is that spectral normalization is applied to every layer in the global version, and only in the output layer for local version. Based on their results, we chose the **Local SN** variant to investigate in this work.

We implemented this method using the spectral normalization implementation in *PyTorch*. This is equivalent to the original description in the paper (Takase et al., 2022) with a $\delta = 1.0$. This method does not have any other hyperparameters.

**Liu-Lipschitz (Liu et al., 2022)**. Originally used to learn a smooth mapping for neural distance fields, such that interpolation and extrapolation of shapes is possible. The method constrains the Lipschitz upper bound of the network, as a learnable parameter $c_i$ per layer. The weights of each network layer are normalized with regards to $c_i$ and the layer's outputs are computed as such

$$
y = \sigma(\hat{W}_i \cdot x + b_i) \qquad \hat{W}_i = normalization(W_i, softplus(c_i)),
\tag{4}
$$

where $\hat{W}_i$ is the normalized weights and $\sigma(.)$ is an activation function. For brevity, we omit the normalization details here and invite the reader to verify the implementation in the original work of Liu et al. (2022). This method also includes a loss function element that minimizes the values of $c_i$ and has the form

$$\mathcal{L}_c = \lambda \prod_i^N softplus(c_i) \tag{5}$$

where $\lambda$ is a hyperparameter to tune, and $N$ is the number of layers in the network, with one $c_i$ per layer. In our work we use $\lambda = 0.000001$ as reported in the experiments of the original paper (Liu et al., 2022). For initializing the values of $c_i$, we decided for the value of 10.0 after some experimentation.

**LipsNet (Song et al., 2023)**. The most recent out of all the methods we investigate. It proposes a network structure called **LipsNet** that replaces a traditional feedforward layer. Specifically, we investigate the variant **LipsNet-L**, whose output is computed as such

$$y = K(x) \cdot \frac{f(x)}{||\nabla f(x)|| + \epsilon}, \tag{6}$$

where $f(x)$ is a regular feedforward layer and $||\nabla f(x)||$ is the 2-norm of the Jacobian matrix relative to the input $x$, $K(x)$ is the Lipschitz value modeled by a feedforward network and also conditioned on $x$, and $\epsilon$ is a small positive value to avoid division by zero.

The authors of **LipsNet** open-sourced an implementation of their method. However, we found a few differences from the original description in their work. Specifically, Song et al. (2023) states that the activation of the $K(x)$ module is a softplus function, but in the open-source code a linear activation is used. In our implementation, we used a softplus activation as in the original paper. Additionally, we opted to not use a $tanh$ squashing function in the outputs of the network and use a linear activation, the same as every other method we experiment. The complete hyperparameters are presented in the Appendix Tables 2, 5, and 9.

## 3   METHOD AND EXPERIMENTAL SETUP

All experiments are run using PPO with a focus in continuous observations and continuous action spaces. We benchmark traditional RL environments using Gymnasium (Towers et al., 2023), and robotics application scenarios with Isaac Gym (Makoviychuk et al., 2021) and deployment in the real-world. The base PPO implementation used are *Stable Baselines* Hill et al. (2018) for the Gymnasium environments and the *RL Games* Makoviichuk & Makoviychuk (2021) implementation for Isaac Gym. Our implementations are written using *PyTorch*.

For every environment and for every method, we trained policies from scratch using 9 different random seeds. Where applicable, we utilized the same hyperparameters for the same environments presented in the original works. Additionally, in Appendix Sections A.1, A.2 and A.3 we present the complete hyperparameters for every method and environment, including the implementation details and reward functions of the locomotion environments introduced in this work.

### 3.1   HYBRID METHODS

We investigate the effectiveness of hybrid methods that combine elements the architectural as well as loss regularization approaches. While we could experiment with every possible combination, we chose to focus on the combination of the **LipsNet** approach with the **L2C2** and **CAPS** regularizations. We decided to exclude **Local SN** due to inferior performance and inferior training stability, and exclude **Liu-Lipschitz** due to the method similarity with **LipsNet** paired with inferior performance. Specifically, we propose and experiment with two hybrid methods: **LipsNet + CAPS**, and **LipsNet + L2C2**.

## 3.2 METRICS

**Cumulative Return**. The cumulative sum of the reward at every step throughout a whole episode $C = \sum_{t=0}^{N} R_t$. It provides a measure of the performance of the policy. This metric is environment dependent, and is used primarily to analyze the trade-off between smoothness and performance.

**Smoothness**. We adopt the same smoothness metric as Mysore et al. (2021) based on frequency spectrum from a Fast Fourier Transform (FFT). The smoothness measure $Sm$ computes a normalized weighted mean frequency and has the form

$$Sm = \frac{2}{n\,f_s} \sum_{i=1}^{n} M_i f_i, \tag{7}$$

where $n$ is the number of frequency bands, $f_s$ the sampling frequency, and, $M_i$ and $f_i$ are the amplitude and frequency of band $i$, respectively. Higher values indicate the presence of high frequency components of large magnitude, and lower values indicate a smoother control signal. In the same manner as the cumulative return, a good smoothness value differs from environment to environment.

## 3.3 EVALUATION SCENARIOS

**Gymnasium Baselines**. Gymnasium Towers et al. (2023) provides standard and classical RL environments for easy and diverse comparisons across different algorithms. We evaluate on 4 continuous control environments: Pendulum-v1, Reacher-v4, LunarLander-v2 (Continuous), and Ant-v4. Because Pendulum-v1 is a simpler environment we train the policies for just 150k timesteps, while the remaining environments train for a total of 400k timesteps. For evaluation, the metrics are computed from 1000 episodes for each training seed.

**Locomotion – Motion Imitation**. Using Isaac Gym (Makoviychuk et al., 2021), we implemented a motion imitation task (Peng et al., 2018; 2020). The quadruped agent is rewarded for matching the reference animation of a pace motion. The agent is trained for a total of 150M timesteps. The evaluation metrics are computed from 50k episodes for each training seed. The details of this environment are presented in the Appendix Section A.2.

**Locomotion – Velocity Controller**. Also with Isaac Gym (Makoviychuk et al., 2021), we train a quadruped agent to perform locomotion while following velocity commands. We employed the same reward objectives as (Rudin et al., 2021), with the exception of action rate penalization which we did not use. The agent is trained for a total of 300M timesteps. The evaluation metrics are computed from 50k episodes for each training seed. The details of this environment are presented in the Appendix Section A.3.

**Locomotion in the Real-World**. For the Motion Imitation and Velocity Controller above, we conduct an experiment with a real-world quadruped robot. First, we identify the best performing method in simulation. Then, we train a policy with the best method and a baseline vanilla policy. To ensure successful sim-to-real transfer these policies are trained with domain randomization (Tobin et al., 2017). The parameters used for training the deployment version of the policies are presented in Table A.2. At deployment time in the real-world, we record 8-second trajectories and disturb the robot by pushing and lifting it off the ground. We compute and report the smoothness $Sm$ of the whole 8-second trajectory.

## 4 EXPERIMENTS AND RESULTS

The main results of our benchmark are condensed in Table 1. Also for reference, the training curves depicting the episode mean reward are depicted in Figure 1.

We can observe that in most environments all methods successfully learn a smoother policy and maintain considerable task performance compared to the **Vanilla** policy. The loss regularization methods **CAPS** and **L2C2** perform similarly in most cases, with a single notable exception in the *Locomotion - Imitation* task, where **CAPS** performed worse than even the **Vanilla** baseline. For the architectural methods, **LipsNet** outperforms **Local SN** and **Liu-Lipschitz** significantly. In the

**Cumulative Return ↑**

| Environment | Vanilla | CAPS
Mysore et al. (2021) | L2C2
Kobayashi (2022) | Local SN
Takase et al. (2022) |
|---|---|---|---|---|
| Pendulum | $-944 \pm 56.8$ | $-940 \pm 50.6$ | $-962 \pm 43.7$ | $-1099 \pm 47$ |
| Ant | $833 \pm 110.2$ | $1027 \pm 134.5$ | $791 \pm 104.1$ | $1108 \pm 174.4$ |
| Reacher | $-6.05 \pm 0.46$ | $\mathbf{-5.98 \pm 0.21}$ | $-6.14 \pm 0.49$ | $-8.73 \pm 0.31$ |
| Lunar | $170 \pm 48.9$ | $-117 \pm 43.4$ | $\mathbf{192 \pm 31.6}$ | $-126 \pm 27.5$ |
| Imitation | $697 \pm 19.3$ | $689 \pm 26.0$ | $\mathbf{697 \pm 16.6}$ | $522 \pm 131.6$ |
| Velocity | $5.98 \pm 0.05$ | $\mathbf{5.99 \pm 0.03}$ | $5.86 \pm 0.04$ | $5.71 \pm 0.19$ |
| **Environment** | **Liu-Lipschitz**
Liu et al. (2022) | **LipsNet**
Song et al. (2023) | **LipsNet + CAPS**
Hybrid (Ours) | **LipsNet + L2C2**
Hybrid (Ours) |
| Pendulum | $-1056 \pm 111.1$ | $-934 \pm 444.7$ | $\mathbf{-737 \pm 181.9}$ | $-870 \pm 171.9$ |
| Ant | $137 \pm 209.5$ | $959 \pm 506.2$ | $\mathbf{1683 \pm 228.4}$ | $\mathbf{1684 \pm 415.2}$ |
| Reacher | $-7.68 \pm 0.86$ | $-6.34 \pm 0.69$ | $-6.13 \pm 0.30$ | $-6.27 \pm 0.44$ |
| Lunar | $92 \pm 69.5$ | $114 \pm 71.2$ | $-304 \pm 22.1$ | $-281 \pm 72.0$ |
| Imitation | $644 \pm 53.1$ | $682 \pm 26.5$ | $673 \pm 26.6$ | $678 \pm 32.9$ |
| Velocity | $5.98 \pm 0.05$ | $5.86 \pm 0.12$ | $5.91 \pm 0.06$ | $5.83 \pm 0.13$ |

**Smoothness $Sm \downarrow$**

| Environment | Vanilla | CAPS
Mysore et al. (2021) | L2C2
Kobayashi (2022) | Local SN
Takase et al. (2022) |
|---|---|---|---|---|
| Pendulum | $0.766 \pm 0.037$ | $0.732 \pm 0.059$ | $0.727 \pm 0.078$ | $0.395 \pm 0.068$ |
| Ant | $1.939 \pm 0.565$ | $1.111 \pm 0.403$ | $1.491 \pm 0.427$ | $\mathbf{0.697 \pm 0.289}$ |
| Reacher | $0.062 \pm 0.108$ | $0.052 \pm 0.005$ | $0.056 \pm 0.009$ | $0.060 \pm 0.011$ |
| Lunar | $0.624 \pm 0.705$ | $0.220 \pm 0.066$ | $0.543 \pm 0.078$ | $0.390 \pm 0.036$ |
| Imitation | $0.679 \pm 0.164$ | $0.697 \pm 0.156$ | $0.524 \pm 0.154$ | $0.631 \pm 0.0603$ |
| Velocity | $0.402 \pm 0.013$ | $0.396 \pm 0.015$ | $0.524 \pm 0.154$ | $0.348 \pm 0.189$ |
| **Environment** | **Liu-Lipschitz**
Liu et al. (2022) | **LipsNet**
Song et al. (2023) | **LipsNet + CAPS**
Hybrid (Ours) | **LipsNet + L2C2**
Hybrid (Ours) |
| Pendulum | $0.492 \pm 0.106$ | $0.944 \pm 0.416$ | $\mathbf{0.314 \pm 0.076}$ | $0.640 \pm 0.284$ |
| Ant | $1.066 \pm 0.278$ | $1.380 \pm 0.357$ | $0.748 \pm 0.095$ | $0.870 \pm 0.167$ |
| Reacher | $0.047 \pm 0.013$ | $0.111 \pm 0.123$ | $\mathbf{0.035 \pm 0.005}$ | $\mathbf{0.035 \pm 0.011}$ |
| Lunar | $0.623 \pm 0.076$ | $0.550 \pm 0.296$ | $\mathbf{0.059 \pm 0.011}$ | $0.095 \pm 0.021$ |
| Imitation | $0.657 \pm 0.101$ | $0.645 \pm 0.132$ | $0.603 \pm 0.120$ | $\mathbf{0.520 \pm 0.067}$ |
| Velocity | $0.402 \pm 0.013$ | $0.298 \pm 0.099$ | $0.275 \pm 0.068$ | $\mathbf{0.255 \pm 0.064}$ |

Table 1: Benchmark of task performance and smoothness of different algorithms in the literature. Each method is trained from scratch with 9 different seeds. The table shows the mean and 1 standard deviation of smoothness and return for 9 seeds. Pendulum, Ant, Reacher and Lunar are Gymnasium environments, and Imitation and Velocity are locomotion tasks in Isaac Gym.

cases that **Local SN** and **Liu-Lipschitz** are smoother than **LipsNet**, it also resulted in reduced task performance, evidenced by the lower mean cumulative return.

The hybrid methods **LipsNet + CAPS** and **LipsNet + L2C2** outperforms the existing methods for nearly every environment. The *Lunar* environment is the single exception where they are clearly inferior. Although the smoothness is better, the cumulative return is too low. We hypothesize that situations like this might happen due to the policy "getting stuck" too early in optimizing for smoothness, rather than task performance. More extensive hyperparameters search and better scheduling of learning rate and loss weights could yield better outcomes. In every other case, Table 1 shows that the hybrids maintain the same level of task performance as the **Vanilla** baseline or even outperform it. It also produces the smoothest policies overall.

For the locomotion tasks, we identify the hybrid **LipsNet + L2C2** as the superior combination, and choose it as the method for deployment in the real-world locomotion tasks. The reuslts of the real-world deployment are presented in Figure 2. The measured smoothness shows that the hybrid method resulted in smoother policies than the **Vanilla** baseline for both locomotion tasks. The

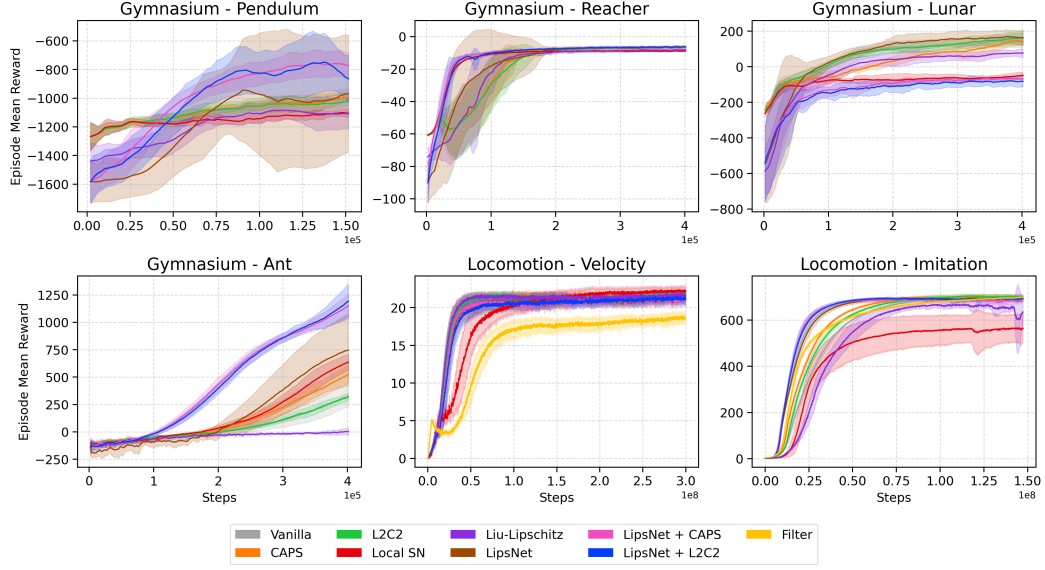

Figure 1: Reward curves during training for 9 seeds. The hybrid methods **LipsNet + CAPS** and **LipsNet + L2C2** show superior or comparable all environments, except in *Lunar*. Note that *Filter* is only employed in the Locomotion tasks.

difference is further magnified in the scenario where the agent is disturbed during task execution. The improvement is easily noticeable when the agent is prevented from executing the task, such as completely lifting the robot in the air. With the **Vanilla** policy the agent generates high-frequency oscillations in sudden bursts, which are undesirable and could even cause hardware damage. On the other hand, the hybrid policy **LipsNet + L2C2** elegantly stops execution until the agent is set to the ground again. A video showcasing this emergent behavior is included in the supplementary material accompanying this paper.

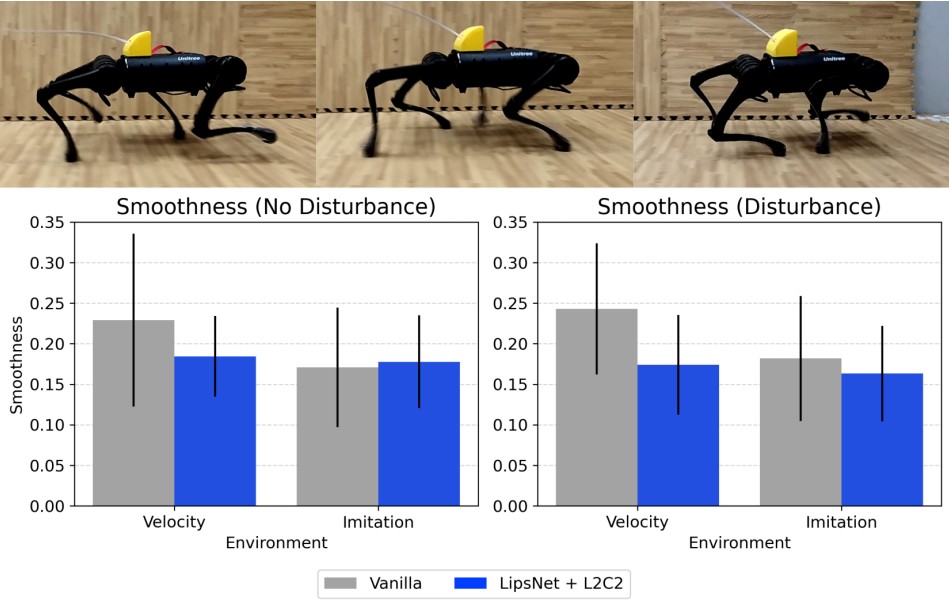

Figure 2: We deploy the best methods from the locomotion tasks to the real-world and measure smoothness. The hybrid method **LipsNet + L2C2** results in smoother policies both under regular execution and when the agent is under disturbances.

## 5 CONCLUSION

In this work we presented a benchmark of methods that can reduce the frequency of oscillations in reinforcement learning policies. We identified existing methods from the literature and classified them according to their mechanism. We proposed two broad categories: loss regularization and architectural methods. Loss regularization methods rely purely on adding regularization elements to the standard policy gradient loss. On the other hand, architectural methods also include introduction and modification of network elements such as weight normalization and special modules that replace feedforward layers. Additionally, by combining a few of the existing works, we also introduced the concept of hybrid methods which have properties from both categories.

Our benchmark included 4 traditional RL environments and 2 robotics locomotion tasks. We analyzed every method in regards of task performance as well as smoothness of output actions. With few exceptions, every method tested performed better than the standard **Vanilla** baseline in every task. Overall, the hybrid method **LipsNet + L2C2** showed the best trade-off between smoothness and task performance, including in the real-world deployment scenario with disturbances. As such, we would recommendation that practitioners that wish to deploy RL policies in the real-world consider training the policies with a hybrid method or another one of the methods investigated here.

In the future, we wish to investigate tasks with an emphasis on more diverse robotics applications. The reduction of high-frequency oscillations is essential for successful sim-to-real transfer and to prevent hardware damage. As such, the community would benefit with a clear set of guidelines on how to train policies for tasks such as locomotion, pick and place, contact rich manipulation, etc.

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

# A    APPENDIX

## A.1    GYMNASIUM ENVIRONMENTS

We experiment with 4 continuous control environments built with Gymnasium (Towers et al., 2023): *Pendulum-v1*, *Reacher-v4*, *LunarLander-v2*, and *Ant-v4*. Table 2 shows the parameters for every method we experiment in our work. Note that we maintain the same hyperparameters of each method for every environment for a fair comparison.

| Method | Parameter | Value |
|---|---|---|
| CAPS | $\sigma$ | 0.1 |
| | $\lambda_T$ | 0.1 |
| | $\lambda_S$ | 0.5 |
| L2C2 | $\sigma$ | 1.0 |
| | $\underline{\lambda}$ | 0.01 |
| | $\overline{\lambda}$ | 1.0 |
| | $\beta$ | 0.1 |
| LipsNet | Weight $\lambda$ | 0.1 |
| | $\epsilon$ | 0.0001 |
| | Initial Lipschitz constant $K_{\text{init}}$ | 1.0 |
| | Hidden layers in $f(x)$ | [64, 64] |
| | Activation in $f(x)$ | ELU |
| | Hidden layers in $K(x)$ | [32] |
| | Activation in $K(x)$ | Tanh |
| Liu-Lipschitz | Weight $\lambda$ | $1 \times 10^{-6}$ |
| | Initial Lipschitz constant | 10.0 |

Table 2: Hyperparameters for the methods investigated in our work for the Gymnasium environments. The same hyperparameters are used for every environment.

## A.2    LOCOMOTION – MOTION IMITATION

The motion imitation policy trains a quadruped robot modeled after the Unitree A1 robot. The agent is tasked with imitating an animation of a forward-moving pace motion. Our implementation uses the same reward elements as Peng et al. (2018) and Peng et al. (2020) and its weights and parameters are presented in Table 3. Table 4 shows the hyperparameters of the base PPO learning algorithm and Table 5 the hyperparameters of the methods investigated in this work. The policy is a 2-layer feed-forward neural network with 512, and 256 hidden units. Each layer uses ELU activations except for the last layer which is linear. The actions are target joint positions for the PD controller with stiffness $k_p = 50.0$ and damping $k_d = 1.2$.

| Reward Term | Value |
|---|---|
| Weight - DoF Position | 0.5 |
| Weight - DoF Velocity | 0.05 |
| Weight - End-effector Position | 0.2 |
| Weight - Root Position | 0.15 |
| Weight - Root Velocity | 0.1 |
| Scale - DoF Position | 5.0 |
| Scale - DoF Velocity | 0.1 |
| Scale - End-effector | 40.0 |
| Scale - Root position | 20.0 |
| Scale - Root velocity | 2.0 |

Table 3: Reward terms for **Locomotion - Motion Imitation** environment.

| Parameter | Value |
|---|---|
| Number of Environments | 4096 |
| Horizon Length | 24 |
| Batch size | 98304 |
| Learning rate | $3 \times 10^{-4}$ |
| Clip rate | 0.2 |
| KL-divergence | 0.008 |
| Entropy coefficient | 0.00 |
| Discount factor | 0.95 |
| GAE discount factor | 0.95 |

Table 4: PPO hyperparameters for training the **Locomotion - Motion Imitation** environment.

| Method | Parameter | Value |
|---|---|---|
| CAPS | $\sigma$ | 0.2 |
| | $\lambda_T$ | 0.01 |
| | $\lambda_S$ | 0.05 |
| L2C2 | $\sigma$ | 1.0 |
| | $\underline{\lambda}$ | 0.01 |
| | $\overline{\lambda}$ | 1.0 |
| | $\beta$ | 0.1 |
| LipsNet | Weight $\lambda$ | $1 \times 10^{-3}$ |
| | $\epsilon$ | 0.0001 |
| | Initial Lipschitz constant $K_{\text{init}}$ | 1.0 |
| | Hidden layers in $f(x)$ | [512, 256] |
| | Activation in $f(x)$ | ELU |
| | Hidden layers in $K(x)$ | [32] |
| | Activation in $K(x)$ | Tanh |
| Liu-Lipschitz | Weight $\lambda$ | $1 \times 10^{-6}$ |
| | Initial Lipschitz constant | 10.0 |
| Butterworth Low-pass Filter | Order | 2 |
| | Cutoff Frequency | 4.0 Hz |

Table 5: Hyperparameters for the methods investigated in our work for the **Locomotion - Motion Imitation** environment.

| Parameter | Value | Type |
|---|---|---|
| Action Noise | 0.02 | Additive |
| Rigid Bodies Mass | [0.95, 1.05] | Scaling |
| Stiffness Gain (PD Controller) | [45, 55] | – |
| Damping Gain (PD Controller) | [0.9, 1.2] | – |
| Ground Friction | [0.1, 1.5] | – |
| Sensor Noise - Orientation | 0.06 | Additive |
| Sensor Noise - Linear Velocity | 0.25 | Additive |
| Sensor Noise - Angular Velocity | 0.3 | Additive |
| Sensor Noise - Joint Angles | 0.02 | Additive |
| Sensor Noise - Feet Contacts | 0.2 | Probability |

Table 6: Domain randomization parameters used to train the policies that are deployed to the real-world.

### A.3 LOCOMOTION – VELOCITY CONTROLLER

We use the Isaac Gym simulator with the legged_gym library (Rudin et al., 2021) for training the velocity controller. The observation consists of the base linear and angular velocities, measurement of the gravity vector, joint positions and velocities, the previous action, binary feet contact, and the velocity command. The reward is a weighted sum of 11 terms listed in Table 7. The policy is a 2-layer feed-forward neural network with both 256 hidden units. Each layer uses ELU activations

except for the linear output layer. The actions are target joint positions for the PD controller with stiffness $k_p = 30$ and damping $k_d = 1.0$. All smoothing methods use the same set of PPO hyper-parameters list in Table 8. In **Filter** method, We added extra states from the past two timesteps to the observation and selected the Butterworth filter as the action filter. Table 9 shows method-specified parameters.

| Reward Term | Weight |
|---|---|
| Linear velocity tracking | 0.02 |
| Angular velocity tracking | 0.01 |
| Linear velocity penalty | $-0.04$ |
| Angular velocity penalty | $-0.001$ |
| Joint acceleration | $-5 \times 10^{-9}$ |
| Joint position limit | $-0.2$ |
| Feet air time | 0.03 |
| Collisions | 0.02 |
| Base height | $-0.2$ |
| Orientation | $-0.02$ |
| Torques | $-4 \times 10^{-6}$ |

Table 7: Reward terms for **Locomotion - Velocity Controller** environment.

| Parameter | Value |
|---|---|
| Batch size | $98304(4096 \times 24)$ |
| Learning rate | $5 \times 10^{-4}$ |
| Clip rate | 0.2 |
| KL-divergence | 0.01 |
| Entropy coefficient | 0.02 |
| Discount factor | 0.99 |
| GAE discount factor | 0.95 |

Table 8: PPO hyper-parameters for training the **Locomotion - Velocity Controller** environment.

| Method | Parameter | Value |
|---|---|---|
| CAPS | $\sigma$ | 0.2 |
| | $\lambda_T$ | 0.01 |
| | $\lambda_S$ | 0.05 |
| L2C2 | $\lambda_\pi$ | 1.0 |
| | $\lambda_V$ | 0.1 |
| LipsNet | Weight $\lambda$ | $1 \times 10^{-3}$ |
| | Initial Lipschitz constant $K_{\text{init}}$ | 1.0 |
| | Hidden layers in $K(x)$ | [32] |
| | Activation in $K(x)$ | Tanh |
| Liu-Lipschitz | Weight $\lambda$ | $1 \times 10^{-6}$ |
| | Initial Lipschitz constant | 10.0 |
| Butterworth Low-pass Filter | Order | 2 |
| | Cutoff Frequency | 4.0 Hz |

Table 9: Method-specific hyper-parameters for **Locomotion - Velocity Controller** environment.

## A.4 ALL RESULTS

In this section we present the results of all three environment types: Gymnasium, Locomotion - Velocity Controller, and Locomotion - Motion Imitation. For every experiment we measured three metrics: mean cumulative rewards (illustrated in Fig. 3, smoothness (depicted in Fig. 4 and action fluctuation (displayed in Fig. 5). Action fluctuation is an alternative metric that can help evaluate the smoothness of a learned policy. It is the mean action difference between two consecutive actions,

where lower values indicating more smoothness and reduced instability. The action fluctuation is computed as follows:

$$\frac{1}{T}\sum_{t=1}^{T}(a_t - a_{t-1})^2, \tag{8}$$

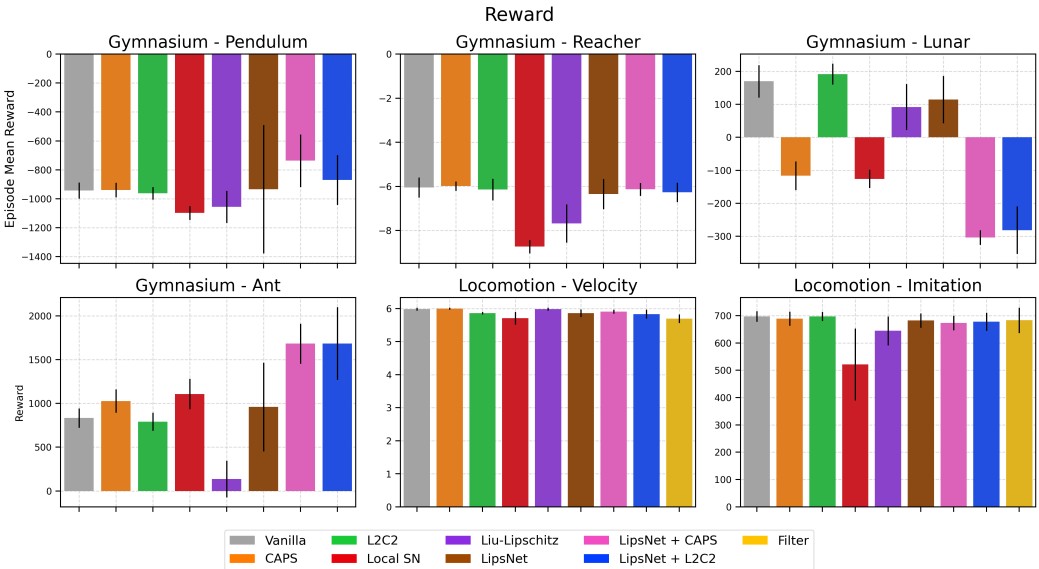

Figure 3: Average cumulative reward for every environment and method.

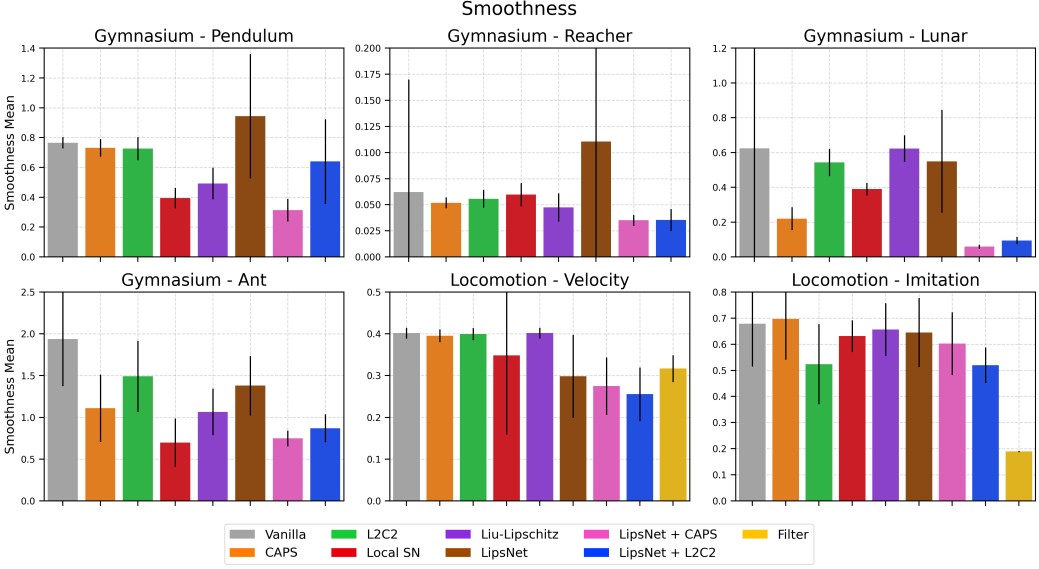

Figure 4: Smoothness evaluation for every environment and method.

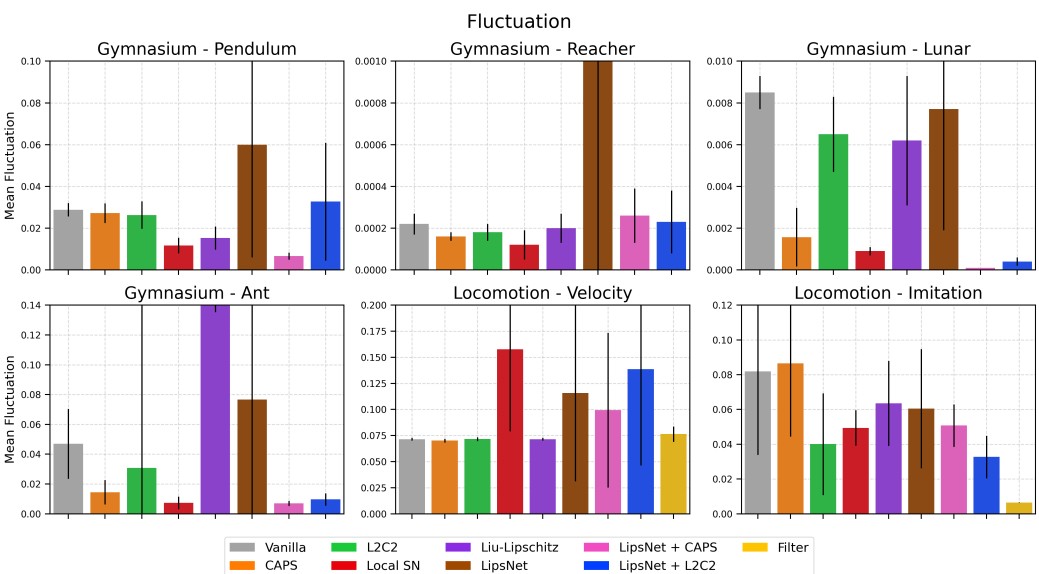

Figure 5: Action fluctuation for every environment and method.

