# OpenReview forum: "Benchmarking Smoothness and Reducing High-Frequency Oscillations in Continuous Control Policies"
_ICLR.cc/2024/Conference — Submitted to ICLR 2024_

### Official Review · Reviewer_5FVX · 2023-10-24

**Soundness:** 2 fair
**Presentation:** 2 fair
**Contribution:** 2 fair
**Rating:** 5
**Confidence:** 5

**Summary:**

The authors investigate methods for reducing the emergence of high-frequency oscillatory behavior when learning continuous control policies. Such behavior is undesirable in real-world robotics applications, however, explicit regularization introduces complex trade-offs. It is then important to identify approaches that gracefully prevent high-frequency motions while maintaining overall performance. This paper aims to provide a benchmark analysis of various method aiming to address this challenge.

**Strengths:**

- Investigating methods to reduce high-frequency oscillatory behavior in continuous control is an important research direction with real-world impact
- PPO is a good primary baseline choice due to its recent success in enabling sim-to-real transfer of complex behaviors in robotics
- Results are averaged over 9 seeds to yield statistical significance

**Weaknesses:**

- The discussion of related work is rather limited and should be expanded by a control theory angle, particularly relating to natural emergence of bang-bang-type controllers [1] or pulse-width modulation. Established benchmarks such as the DeepMind Control Suite can furthermore often be solved with high-frequency bang-bang control as studied in [2] or general discretization in [3] – these works provide further motivation for the need of well-designed benchmarks as this paper aims to provide.
- A very relevant related work that automatically learns regularization trade-offs was presented in [4]
- The majority of cumulative returns as well as smoothness scores in Table 1 have overlapping error bands making performance differences difficult to judge.
- The current evaluation has insufficient breadth to serve as a benchmark for smooth continuous control. The analysis relies on empirical data and as such should cover a large representative range of (robotics) tasks or baseline algorithms, ideally both.
- Pendulum/Reacher/Lunar Lander are rather toy-ish examples that are not representative of the challenges the paper aims to analyze, while they could provide good illustrative examples if analyzed more in-depth
- The analysis would profit from extension to a broader set of baseline algorithms other than PPO to identify trends in order to serve as an extensive benchmark



[1] R. Bellman, et al. "On the “bang-bang” control problem," Quarterly of Applied Mathematics, 1956.

[2] T. Seyde, et al. "Is Bang-Bang control All You Need? Solving Continuous Control with Bernoulli Policies," NeurIPS, 2021.

[3] Y. Tang, et al. "Discretizing Continuous Action Space for On-Policy Optimization," AAAI, 2020.

[4] S. Bohez, et al. "Value constrained model-free continuous control," arXiv, 2019.

**Questions:**

- Why did the “Locomotion – Velocity Controller” task not use the action rate penalization of Rudin (2021)?
- What do state and action trajectories of trained policies look like? How do they compare across agents?
- How does policy performance and smoothness compare across agents when simply varying weights of action magnitude/smoothness penalties in the reward function? As this is commonly how practitioners counteract high-frequency behavior, such an analysis would improve insights drawn from a benchmark.

---

> ### Author Response · Authors · 2023-11-22
>
> We thank the reviewer for his time and for providing feedback to improve our work.
>
> > The discussion of related work is rather limited and should be expanded by a control theory angle.
>
> The related works section of our paper lists other benchmark papers and focuses mainly on categorizing and describing past methods that learns policies that produce smooth control behaviors, with few oscillations. Some of the methods investigated in our work do have more of a control theory perspective (esp. [1] and [2]), though not all of them. For example, Liu-Lipschitz [3] is an approach originally presented in the context of 3D imaging.
>
> Because in our work we are interested in measuring the effectiveness of these approaches in a performance x smoothness analysis more so than their mechanisms, we believe an extended discussion from a control theory angle is a bit out of scope. We would invite an interested reader to check the original papers for details, specially for methods proposed from a control perspective.
>
> > Established benchmarks such as the DeepMind Control Suite can furthermore often be solved with high-frequency bang-bang control as studied in [...] or general discretization in [...]
>
> Thank you for the reference suggestions. Although these works demonstrate that bang-bang control and discretization can solve the tasks, we did not include these types of controllers in our benchmark since they are the opposite of the type of smooth control we wish to achieve. A major part of our benchmark is done with real-world hardware and bang-bang/discrete type controllers cannot be safely deployed.
>
> Nonetheless, the works listed are interesting benchmarks that evaluate performance in many environments, and we'll include them in the discussion of related works in the final paper.
>
> > A very relevant related work that automatically learns regularization trade-offs was presented in [4]
>
> We apologize for missing the related work. The work cited by the reviewer is indeed very relevant. We'll include it in the related works section discussion in the final paper version.
>
> > The majority of cumulative returns as well as smoothness scores in Table 1 have overlapping error bands making performance differences difficult to judge.
>
> We argue that our benchmark results show a significant improvement over a "Vanilla" policy versus regularization methods, showing that regularization is indeed effective. It is true that in a single scenario the smoothness between two regularization techniques might not be significant (see Hybrid methods in "Reacher", "ShadowHand", and "Velocity"). However, analyzing the smoothness x performance  across all scenarios we can observe that the "Hybrid" methods outperformed others more often than not. While other approaches have significant deviations scenario to scenario, it is clear that the "Hybrids" are the most consistent method across the board.
>
> > The current evaluation has insufficient breadth to serve as a benchmark for smooth continuous control.
> > Pendulum/Reacher/Lunar Lander are rather toy-ish examples that are not representative of the challenges the paper aims to analyze
>
> We've added two additional scenarios: "ShadowHand" and "Handstand". These are challenging tasks that require agile and dexterous movements. The Handstand task is also deployed and evaluated with real hardware.
>
> Additionally, we have also measured the performance of every single method in the real-world and included an ablation case of a Vanilla policy trained without domain randomization. Please refer to the comment "Extended Experiments and Results -- Updated Table 1" for the updated version of our results.
>
> > The analysis would profit from extension to a broader set of baseline algorithms other than PPO to identify trends in order to serve as an extensive benchmark
>
> We agree with the reviewer that more extensive and broader experiments are better. Still, PPO is one of the most popular algorithm used by the community, specially for sim2real deployment. We believe the scope of the work is wide enough to be relevant to the research community.
>
> ---
>
> We also invite the reviewer to check the "Summary of Changes" comment for a more extensive list of major and minor changes we've done thanks to your and other reviewers feedback.
>
> ---
>
> [1] - Taisuke Kobayashi. L2c2: Locally lipschitz continuous constraint towards stable and smooth reinforcement learning. In 2022 IEEE/RSJ International Conference on Intelligent Robots and Systems (IROS), pp. 4032–4039. IEEE, 2022
>
> [2] - Ryoichi Takase, Nobuyuki Yoshikawa, Toshisada Mariyama, and Takeshi Tsuchiya. Stability certified reinforcement learning control via spectral normalization. Machine Learning with Applications, 10:100409, 2022.
>
> [3] Hsueh-Ti Derek Liu, Francis Williams, Alec Jacobson, Sanja Fidler, and Or Litany. Learning smooth neural functions via lipschitz regularization. In ACM SIGGRAPH 2022 Conference Proceedings, pp. 1–13, 2022.

---

> > ### Author Response · Authors · 2023-11-22
> >
> > >Why did the “Locomotion – Velocity Controller” task not use the action rate penalization of Rudin (2021)?
> > > How does policy performance and smoothness compare across agents when simply varying weights of action magnitude/smoothness penalties in the reward function? As this is commonly how practitioners counteract high-frequency behavior, such an analysis would improve insights drawn from a benchmark.
> >
> > As the reviewer mentioned, adding an action penalty term to the reward function is a popular practice but it is not "free". Reward functions in RL are known to be difficult to tune, and the more terms added to the function the more challenging of a balancing act it becomes. In an idealized setting, we envision a practitioner designing a reward function with no concern for smoothness, only performance. Once the desired task reward is complete, a smoothness regularization method -- such as the ones investigated in this work -- can be added on top. This process would facilitate reward design and reduce complexity, while guaranteeing smooth control.
> >
> > As such, in the original scenarios evaluated in our work we opted to not use a direct action penalty term in the reward function. Note that "Velocity" and "Handstand" tasks do include reward terms for joint acceleration and joint torques, which can be seen as analogous to action penalty.
> >
> > To address the concerns of the reviewer and also consider the case of a policy with action penalty, we included it for the "Handstand" scenario that was added during the rebuttal process.  Even in this case, the results show that the regularization methods further improve the smoothness compared to a Vanilla policy. (Please refer to the updated Table and Real World experiments in the summary comment.) In conclusion, our results show that whether a direct action penalty term or "indirect" terms (joint acceleration and torque) are employed, regularization is still beneficial and significantly improves smoothness compared to no regularization.
> >
> > > What do state and action trajectories of trained policies look like? How do they compare across agents?
> >
> > Here are trajectory samples of actions and joint angles for "Vanilla" and "Hybrid - LipsNet + L2C2" for the Handstand scenario.
> >
> > * Handstand | Vanilla | Actions: https://i.imgur.com/4GNZmLv.png
> > * Handstand | Hybrid - LipsNet + L2C2 | Actions: https://i.imgur.com/8GI9Wjh.png
> > * Handstand | Vanilla | Joint Angles: https://i.imgur.com/c2JERaq.png
> > * Handstand | Hybrid - LipsNet + L2C2 | Joint Angles: https://i.imgur.com/iqWGMmb.png
> >
> > There is an observable difference between vanilla and the regularized methods. The difference between any two regularized methods is less significant, except in the cases where the method failed to learn a good policy.
> >
> > We'll include an extended comparison of joint angles and actions for every method in a dedicated appendix section in the final paper.

---

> > > ### Comment · Reviewer_5FVX · 2023-11-22
> > > **Response to rebuttal**
> > >
> > > Thank you for your detailed responses and additional evaluation. The new experiments improve the quality of the manuscript and I'll raise my score. However, I still believe that overlapping error bands and small differences in mean performance make improvements difficult to judge, while analysis is not extensive enough from a benchmarking perspective.

---

### Official Review · Reviewer_qSpQ · 2023-11-01

**Soundness:** 3 good
**Presentation:** 3 good
**Contribution:** 2 fair
**Rating:** 6
**Confidence:** 3

**Summary:**

The authors benchmark several recent methods for imposing smoothness on policies. The benchmarks are mainly done in simulation, and one the best performing method is evaluated in a sim2real transfer against a vanilla RL policy.

**Strengths:**

- Benchmark papers are important for the community, and the sim benchmark evaluation on policy smoothness appears to be carefully constructed
- Reducing oscillations is important, and a common problem in sim2real as the authors note in their conclusions
- The paper is mostly well-written and easy to read

**Weaknesses:**

Overall I think this is a useful benchmark paper but it has some issues with framing:
- The authors (and title even) mix the concepts of policy smoothness and reducing high-frequency oscillations, but how these concepts relate and why you get oscillations is not thoroughly defined. The paper also talks about sim2real where oscillations are indeed a common problem, but the authors do not make a convincing argument that this is due to non-smooth policies. Sometimes it might be but this has not been well explored as far as I am aware. Overall the paper does a good job of benchmarking existing approaches to increase policy smoothness, but the sim2real / oscillation connection seems weak.
- More in that vein, the sim2real is only attempted with the best performing method from simulation. If simulation was a good indicator of real-world performance, sim2real would be much easier than it currently is, but this is not the case in my experience. If you want to make this paper about sim2real instead of just policy smoothness, then it would be useful if you had tested all the approaches in the real world to see how the approaches generalized. It would also have been useful to see a video of the experiments as is conventional in robotics. Did the robot actually oscillate or is your smoothness metric just picking up a quick motion (e.g, a simple step function also has high-frequency components)? These can sometimes be desirable.

As it currently stands, I would consider toning down the sim2real implications a little bit and focus more on smoothness.

Minor:
typo: hyperaparameter

**Questions:**

"... but in the open-source code a linear activation is used. In our implementation, we used a softplus activation as in the original paper." - This seems like an arbitrary choice that might degrade the performance of a defacto available option. Maybe the implementation is more up to date, or there was a typo in the paper?  Can you test and confirm that it performs as well, or why not include both versions?

---

> ### Author Response · Authors · 2023-11-21
>
> We thank the reviewer for his time and for providing feedback to improve our work.
>
> > The authors (and title even) mix the concepts of policy smoothness and reducing high-frequency oscillations, but how these concepts relate and why you get oscillations is not thoroughly defined.
>
> The reviewer raises a good point that our language in the paper is ambiguous. When we use the term "policy smoothness" or a "smooth policy" we simply mean a policy that exhibits/produces smooth control behaviors in the form of reduced oscillations, and not necessarily a policy that produces a smooth mapping function. However, and interestingly, the two do seem to be correlated as most methods that we investigated/refer in our paper do use the latter definition of smooth policy and aim to achieve smoothness, as in "a smooth mapping function", which in turns reduces oscillations, evidenced by our results.
>
> We will clarify the meaning of these terms in the paper and better define "smoothness".
>
> >  it would be useful if you had tested all the approaches in the real world to see how the approaches generalized.
>
> We've followed up on the reviewer's suggestions and reran all of our real-world experiments with every method reported on the paper. Additionally, we've also measured the cumulative return in the real-world to enable a performance vs smoothness analysis.
>
> **Updated Real-World Results** - Comparison of All Methods + No DR Ablation + Cumulative Return Computation:
>
> Imitation: https://i.imgur.com/zuWvjqq.png
>
> Velocity: https://i.imgur.com/UG2dvYF.png
>
> Handstand: https://i.imgur.com/vmjOltq.png
> * We did not deploy "No DR" ablation for handstand out of concern for robot and actuator damage.
>
> > It would also have been useful to see a video of the experiments as is conventional in robotics. Did the robot actually oscillate or is your smoothness metric just picking up a quick motion
>
> Our original video demonstrates a case of an emergent behavior for the Hybrid policy where when the robot is picked up from the ground it will slowly stop moving. In contrast, when running the vanilla policy an out of distribution state such as this generates "wild movement" and jerking motions. We've also updated the supplementary material to include demonstrations of a new handstand policy. We can observe in the video that with our proposed hybrid method the motion is smoother during regular execution as well as when disturbed.
>
> > (On a LipsNet modification) This seems like an arbitrary choice that might degrade the performance of a defacto available option. Maybe the implementation is more up to date, or there was a typo in the paper? Can you test and confirm that it performs as well, or why not include both versions?
>
> We've confirmed with the original authors of LipsNet that the version we implemented is indeed the correct and intended way. The authors had incorrectly used a linear activation as a default when open-sourcing their code (The output $K$ of LipsNet must be $ > 0 $. We've confirmed that the activation function used by the authors in their original experiments matches the version we've used in our work. Additionally, the original authors have since updated their open-source code to match the version we've used after our inquiry.
>
> ---
>
> We also invite the reviewer to check the "Summary of Changes" comment for a more extensive list of major and minor changes we've done thanks to your and other reviewers feedback.

---

> > ### Comment · Reviewer_qSpQ · 2023-11-22
> > **Response to authors**
> >
> > I thank the authors for the additional sim2real experiments. I note that pure l2c2 seems pretty close to the results of the hybrid methods on that, especially if you consider both smoothness and returns. I wonder if this is due to low sample size or just because the real world is hard to model?
> >
> > Taken together you can see a trend of sorts in the results, but as another reviewer mentioned the confidence intervals are rather large (as is often the case in RL). I am not sure if it is feasible to get more samples on the sim experiments, but one idea to reduce the variance might be to examine the difference across environment seeds (like a pair-wise test) if you are re-using the same environment seeds. Another would be to normalize and pool all task results for sim and real (respectively) or perhaps put all sim vs. real experiments into a Pareto plot (norm. return vs. smoothness) .
> >
> > Regarding the definition of smoothness (smooth policy -> smooth behavior), I agree that you can make this assumption which also seems common in related work, but it is a rather strong simplification. The final behavior of the system depends not only on the control policy, but also on the system (plant) dynamics. More attention could have been paid to the assumptions being made on the system here. For example, I suspect you might need some assumptions of smoothness of the system dynamics for this to hold. Another common source of oscillations in sim2real is also latency, which is also not mentioned. That said, I still think this can have merit just as a benchmark paper. I hesitate if that part is conclusive and deep enough for an 8 though (see above).

---

### Official Review · Reviewer_RjiF · 2023-11-07

**Soundness:** 3 good
**Presentation:** 3 good
**Contribution:** 2 fair
**Rating:** 5
**Confidence:** 3

**Summary:**

In the paper, the author investigates algorithms aiming to prevent high-frequency oscillations during the RL sim2real transfer process. They conduct a comprehensive benchmark of both the performance and smoothness of the trained policies. Moreover, they devise a hybrid method, which, as per the results, demonstrates superior performance.

**Strengths:**

1. The author effectively categorizes various smoothing methods, providing an exhaustive understanding of the diverse kinds of policy smoothing techniques.
2. Extensive benchmarks are conducted, further enhancing comprehension of the performance exhibited by differing algorithms in simulation.

**Weaknesses:**

1. The paper presents a rather incremental contribution. The author proposes a hybrid method that combines architectural methods and regularization techniques, but it remains unclear how these two components interact or how the policy can be further improved.
2. The performance enhancement of the hybrid method, compared to the baseline, appears limited. Moreover, since oscillations occur during the sim2real process, the author only contrasts the hybrid method with the vanilla policy. This means comparisons to other baselines are omitted, making it unclear how the hybrid method measures up against other methodologies.

**Questions:**

1. In Figure 2, in the disturbance imitation task test, why does the combination of LipsNet + L2C2 underperform compared to the vanilla policy regarding smoothness?
2. As you have exclusively provided the measure of policy smoothness in real-world situations, I'm curious as to how the real-world performance (reward) measures up against other baselines?

---

> ### Author Response · Authors · 2023-11-21
>
> We thank the reviewer for his time and for providing feedback to improve our work.
>
> > As you have exclusively provided the measure of policy smoothness in real-world situations, I'm curious as to how the real-world performance (reward) measures up against other baselines?
>
> We've followed up on the reviewer's suggestions and reran all of our real-world experiments with every method reported on the paper. Additionally, we've also measured the cumulative return in the real-world to enable a performance vs smoothness analysis. (See below).
>
> > In Figure 2, in the disturbance imitation task test, why does the combination of LipsNet + L2C2 underperform compared to the vanilla policy regarding smoothness?
>
> Every policy deployed to the real-world is trained with domain randomization, including environment physics, terrain and actuator properties. The addition of domain randomization by itself results in smoother control behaviors. For a given task, regular execution (i.e. no disturbances) might be smooth enough that there are no significant differences between a regularized policy and a vanilla DR policy. However, when the policy needs to execute a sort of recovery behavior from an unstable state (which could be out of distribution), for example from an external disturbance, we can observe that a policy trained with smoothness regularization is indeed smoother.
>
> For supporting evidence to our point above, we have included an ablation of a "Vanilla (No DR)" vs "Vanilla" which demonstrates this behavior.
>
> ---
> **Updated Real-World Results** - Comparison of All Methods + No DR Ablation + Cumulative Return Computation:
>
> Imitation: https://i.imgur.com/zuWvjqq.png
>
> Velocity: https://i.imgur.com/UG2dvYF.png
>
> Handstand: https://i.imgur.com/vmjOltq.png
> * We did not deploy "No DR" ablation for handstand out of concern for robot and actuator damage.
> ---
>
> We also invite the reviewer to check the "Summary of Changes" comment for a more extensive list of major and minor changes we've done thanks to your and other reviewers feedback.

---

### Official Review · Reviewer_ALp7 · 2023-11-09

**Soundness:** 3 good
**Presentation:** 3 good
**Contribution:** 3 good
**Rating:** 5
**Confidence:** 5

**Summary:**

The paper addresses the issue of high-frequency oscillations in reinforcement learning policies, especially when applied to real-world hardware. The authors categorize methods to mitigate these oscillations into loss regularization and architectural methods, aiming to smooth the input-output mapping of policies. They benchmark these methods on classic RL environments and robotics locomotion tasks, introducing hybrid methods that combine both approaches. The study finds that hybrid methods, particularly LipsNet combined with CAPS and L2C2, perform well in both simulations and real-world tasks, achieving smoother policies without compromising task performance.

**Strengths:**

A technically sound paper with theoretical analysis and experimental support of their discoveries. Aims to get a better understanding and address an important problem in robotics, especially in application to sim2real when high-frequency, noisy policies can damage robot hardware. In addition to sim sim-only results demonstrate the advantages of their approach on a quadruped robot.

**Weaknesses:**

Lack of experiments, especially for more challenging continuous control tasks. In addition would be good to see more detailed comparisons against MLP baseline - see training curves and comparison of reward and variance vs not only a number of samples but also training time.

**Questions:**

1) Could you share training plots at least for Ant and quadruped robot for the reward vs wall-clock time for LipsNet + CAPS and LipsNet + L2C2 vs vanilla MLP? Are there losses in training (wall-clock time) performance when using these more advanced methods vs MLP or they are computationally comparable to the vanilla MLP?
2) Could you run experiments for more challenging control problems - humanoid, or on of the Allegro (Shadow) Hand dexterous manipulation tasks?

---

> ### Author Response · Authors · 2023-11-21
>
> We thank the reviewer for his time and for providing feedback to improve our work.
>
> > Could you share training plots at least for Ant and quadruped robot for the reward vs wall-clock time for LipsNet + CAPS and LipsNet + L2C2 vs vanilla MLP?
>
> Unfortunately, because we ran our experiments in parallel, sharing computing load with other processes and experiments, the wall-clock vs reward graph is not truly representative of performance efficiency. We'll consider running a separate experiment sequentially to accurately measure reward vs (real) time.
>
> > Are there losses in training (wall-clock time) performance when using these more advanced methods vs MLP or they are computationally comparable to the vanilla MLP?
>
> For pure loss-based methods we have not observed any difference in training time. In regards to the architectural methods, Spectral Normalization (SN-Local) and Liu-Lipschitz are both similar to a vanilla MLP in training time. Because our policy networks are not large, the training bottleneck ends up in the sample collection time in this case.
>
> However, it should be noted that the LipsNet method comes with a significant slowdown. We observe an iteration time approximately 10 times longer than other methods, which is line with what the original authors reported in Appendix D. in [1]. By extension, this also affects the training time for the Hybrid methods that we proposed in our paper.
>
> Overcoming such a slowdown and achieving the same smoothness and performance metrics we measured here could be a fruitful path for future research.
>
> We'll incorporate the points above in the discussion in Section 4 of the paper.
>
> > Could you run experiments for more challenging control problems - humanoid, or on of the Allegro (Shadow) Hand dexterous manipulation tasks?
>
> Thank you for your suggestion. We have run our experiments with the ShadowHand task in Isaac Gym. Please refer to the full updated table and comments in our top official comment "Extended Experiments and Results -- Updated Table 1".
>
> We also invite the reviewer to check the "Summary of Changes" comment for a more extensive list of major and minor changes we've done thanks to your feedback.
>
> ---
>
> [1] - Xujie Song, Jingliang Duan, Wenxuan Wang, Shengbo Eben Li, Chen Chen, Bo Cheng, Bo Zhang, Junqing Wei, and Xiaoming Simon Wang. Lipsnet: A smooth and robust neural network with adaptive lipschitz constant for high accuracy optimal control. 2023.

---

### Author Response · Authors · 2023-11-21
**Summary of Changes**

Firstly, we wish to thank all reviewers for taking the time to review our paper and for providing valuable feedback. We have read every review and carefully considered how we can improve our work. The following comments summarize the major changes we've done as of this rebuttal period.

We have extended the experiments and strengthned the results of this work, specifically:
* **We have included two additional challenging control scenarios.**
    * **ShadowHand** - As suggested by Reviewer alp7. This is a challenging problem that involves dexterous manipulation of an object with an articulated robotics hand. The agent is rewarded for matching the pose of a cube object to a randomized target pose. This experiment is done only in simulation.
    * **Handstand** - A challenging balancing problem. The agent is rewarded for balancing on its hind legs and remaining in an upright orientation. Because the robot is not statically stable in a biped stance, it must continuously move to maintain balance, making it an ideal challenging task to measure smoothness. Besides matching an upright orientation, the reward function also includes terms to minimize joint torque and joint acceleration. This task is evaluated both in simulation and with the real-robot.
    * Same as the scenarios originally presented on the paper, we trained 9 independent seeds for each new method. The details of both tasks will be added to the paper to "Section 3.3 Evaluation Scenarios" and dedicated appendix sections for training and reward function details.
    * **In a comment below we present the updated Table 1 that includes the new experiments and will be in the final paper version.**
* **We have conducted an experiment where we evaluate every method in the real world for Imitation, Velocity and Hanstand scenarios.** Additionally, we have included an ablation where we deploy a vanilla policy trained without any domain randomization (DR).
    * The resulting figures that will substitute Figure 2 in the paper are presented in a dedicated comment below.

**Update 1** : We've updated the supplementary material video to include a demonstration of the handstand. It showcases the effect of a Vanilla policy vs our proposed hybrid method LipsNet + L2C2.

---

> ### Author Response · Authors · 2023-11-21
> **Extended Experiments and Results -- Updated Table 1**
>
> # Extended Experiments and Results -- Updated Table 1
>
> **Cumulative Return (Higher is Better)**
> | **Environment**         |  **Pendulum**   |     **Ant**     | **Reacher**        |     **Lunar**     |
> | ----------------------- |:---------------:|:---------------:| ------------------ |:-----------------:|
> | Vanilla                 |   -944 +- 57    |   833 +- 110    | -6\.05 +- 0.46     |     170 +- 49     |
> | CAPS – [1]              |   -940 +- 51    |   1027 +- 135   | **-5\.98 +- 0.21** |    -117 +- 43     |
> | L2C2 – [2]              |   -962 +- 44    |   791 +- 104    | -6\.14 +- 0.49     |   **192 +- 32**   |
> | Local SN – [3]          |   -1099 +- 47   |   1108 +- 174   | -8\.73 +- 0.31     |    -126 +- 28     |
> | Liu-Lipschitz – [4]     |  -1056 +- 111   |   137 +- 210    | -7\.68 +- 0.86     |     92 +- 70      |
> | LipsNet – [5]           |   -934 +- 445   |   959 +- 506    | -6\.34 +- 0.69     |     114 +- 71     |
> | LipsNet + CAPS – Hybrid | **-737 +- 182** | **1683 +- 228** | -6\.13 +- 0.30     |    -304 +- 22     |
> | LipsNet + L2C2 – Hybrid |   -870 +- 172   |   1684 +- 415   | -6\.27 +- 0.44     |    -281 +- 72     |
> | **Environment**         | **ShadowHand**  |  **Imitation**  | **Velocity**       |   **Handstand**   |
> | Vanilla                 |   5025 +- 460   |    697 +- 19    | 5\.98 +- 0.05      |   3\.37 +- 0.10   |
> | CAPS – [1]              |   4421 +- 576   |    689 +- 26    | **5\.99 +- 0.03**  |   3\.35 +- 0.06   |
> | L2C2 – [2]              | **5190 +- 390** |  **697 +- 17**  | 5\.86 +- 0.04      |   3\.41 +- 0.06   |
> | Local SN – [3]          |    166 +- 93    |   522 +- 132    | 5\.71 +- 0.19      |   3\.37 +- 0.07   |
> | Liu-Lipschitz – [4]     |   1213 +- 538   |    644 +- 53    | 5\.98 +- 0.05      |   3\.38 +- 0.03   |
> | LipsNet – [5]           |   4784 +- 333   |    682 +- 27    | 5\.86 +- 0.12      |   3\.40 +- 0.06   |
> | LipsNet + CAPS – Hybrid |   3923 +- 347   |    673 +- 27    | 5\.91 +- 0.06      |   3\.40 +- 0.06   |
> | LipsNet + L2C2 – Hybrid |   4913 +- 305   |    678 +- 33    | 5\.83 +- 0.13      | **3\.45 +- 0.05** |
>
> **Smoothness (Lower is Better)**
> | Environment             |     Pendulum      | Ant               |  Reacher * 10^1   |   Lunar * 10^1    |
> | ----------------------- |:-----------------:| ----------------- |:-----------------:|:-----------------:|
> | Vanilla                 |   0\.77 +- 0.04   | 1\.94 +- 0.57     |   0\.62 +- 1.08   |   6\.24 +- 7.05   |
> | CAPS – [1]              |   0\.73 +- 0.06   | 1\.11 +- 0.40     |   0\.52 +- 0.05   |   2\.20 +- 0.66   |
> | L2C2 – [2]              |   0\.73 +- 0.08   | 1\.49 +- 0.43     |   0\.56 +- 0.09   |   5\.43 +- 0.78   |
> | Local SN – [3]          |   0\.40 +- 0.07   | **0\.70 +- 0.29** |   0\.60 +- 0.11   |   3\.90 +- 0.36   |
> | Liu-Lipschitz – [4]     |   0\.49 +- 0.11   | 1\.07 +- 0.28     |   0\.47 +- 0.13   |   6\.23 +- 0.76   |
> | LipsNet – [5]           |   0\.94 +- 0.42   | 1\.38 +- 0.36     |   1\.11 +- 1.23   |   5\.50 +- 2.96   |
> | LipsNet + CAPS – Hybrid | **0\.31 +- 0.08** | 0\.75 +- 0.10     | **0\.35 +- 0.05** | **0\.59 +- 0.11** |
> | LipsNet + L2C2 – Hybrid |   0\.64 +- 0.28   | 0\.87 +- 0.17     |   0\.35 +- 0.11   |   0\.95 +- 0.21   |
> | **Environment**         |  **ShadowHand**   | **Imitation**     |   **Velocity**    |   **Handstand**   |
> | Vanilla                 |   1.82 +- 0.09    | 0\.68 +- 0.16     |   0\.40 +- 0.01   |   0\.64 +- 0.11   |
> | CAPS – [1]              |   1.63 +- 0.15    | 0\.70 +- 0.16     |   0\.40 +- 0.02   |   0\.63 +- 0.07   |
> | L2C2 – [2]              |   1.64 +- 0.07    | 0\.52 +- 0.15     |   0\.52 +- 0.15   |   0\.56 +- 0.05   |
> | Local SN – [3]          | 0.09 +- 0.08 (*)  | 0\.63 +- 0.06     |   0\.35 +- 0.19   |   0\.58 +- 0.04   |
> | Liu-Lipschitz – [4]     |  1.28 +- 0.2 (*)  | 0\.66 +- 0.10     |   0\.40 +- 0.01   |   0\.66 +- 0.06   |
> | LipsNet – [5]           |   1.77 +- 0.07    | 0\.65 +- 0.13     |   0\.30 +- 0.10   |   0\.55 +- 0.03   |
> | LipsNet + CAPS – Hybrid | **1.58 +- 0.06**  | 0\.60 +- 0.12     |   0\.28 +- 0.07   |   0\.56 +- 0.06   |
> | LipsNet + L2C2 – Hybrid | **1.57 +- 0.08**  | **0\.52 +- 0.07** | **0\.26 +- 0.06** | **0\.46 +- 0.04** |
>
> \(*) Low smoothness but subpar performance.
>
> * We transposed the Table for better reading of our results.
> * The two newly included scenarios further reinforce the benefit of the smoothing methods compared to nothing/Vanilla. Additionally, the results also follow a similar pattern with the hybrid methods outperforming every other method in smoothness while maintaining a high performance (cumulative return).

---

> ### Author Response · Authors · 2023-11-21
> **Extended Experiments and Results - Real-World**
>
> # Extended Experiments and Results - Real-World
>
> * At the request of multiple reviewers we've greatly extended the breadth of our experiments particularly in the real-world. Where originally we only deployed the best performing method from simulation, we have now ran and benchmarked every method presented in the paper for scenarios where sim2real is applicable.
> * **Included a new challenging real-world scenario: handstand**. The agent is rewarded for balancing on its hind legs and remaining upright.
> * **Ran all 8 methods for the 3 real-world scenarios**.
> * **Included an ablation for a policy trained without domain randomization (DR)**.
> * **Measured the real-world performance for each task with an alternative reward function**.
>     * The reward function in the real-world is representative of the task and contains similar components to the simulation function. The details will be added to a dedicated appendix section.
> * Compiled the results in bar graphs that will supersede Figure 2.
>     * Imitation: https://i.imgur.com/zuWvjqq.png
>     * Velocity: https://i.imgur.com/UG2dvYF.png
>     * Handstand: https://i.imgur.com/vmjOltq.png
> * Discussion:
>     * The results show that "Hybrid - LipsNet + L2C2" was the best method. It is not the only method that achieves a good combination of performance and smoothness, but it remained consistent for every tested scenario, while other methods faltered.
>     * The domain randomization (DR) ablation shows that in addition to (greatly) improving real-world performance DR also results in smoother behaviors, see "Vanilla" vs "Vanilla (No DR)". Note that every method was trained with DR for the real-world experiment test. Still, the inclusion of architectural or loss-based methods as we've studied on this paper can further improve smoothness while maintaining comparable task performance.
>     * Note that the "No DR" ablation was not deployed for the handstand scenario. This is a challenging scenario with unstable states and potential for robot collapse and actuator damage. We believe that the ablation for the other two scenarios is sufficient evidence that DR by itself can result in smoother control.
> * The discussion points above will be added to "Section 4. Experiments and Results" along with the graph to replace Figure 2.
>
> ### References
> [1] - Siddharth Mysore, Bassel Mabsout, Renato Mancuso, and Kate Saenko. Regularizing action policies for smooth control with reinforcement learning. In 2021 IEEE International Conference on Robotics and Automation (ICRA), pp. 1810–1816. IEEE, 2021.
>
> [2] - Taisuke Kobayashi. L2c2: Locally lipschitz continuous constraint towards stable and smooth reinforcement learning. In 2022 IEEE/RSJ International Conference on Intelligent Robots and Systems (IROS), pp. 4032–4039. IEEE, 2022
>
> [3] - Ryoichi Takase, Nobuyuki Yoshikawa, Toshisada Mariyama, and Takeshi Tsuchiya. Stability certified reinforcement learning control via spectral normalization. Machine Learning with Applications, 10:100409, 2022.
>
> [4] - Hsueh-Ti Derek Liu, Francis Williams, Alec Jacobson, Sanja Fidler, and Or Litany. Learning smooth neural functions via lipschitz regularization. In ACM SIGGRAPH 2022 Conference Proceedings, pp. 1–13, 2022.
>
> [5] - Xujie Song, Jingliang Duan, Wenxuan Wang, Shengbo Eben Li, Chen Chen, Bo Cheng, Bo Zhang, Junqing Wei, and Xiaoming Simon Wang. Lipsnet: A smooth and robust neural network with adaptive lipschitz constant for high accuracy optimal control. 2023.

---

### Meta-Review · Area_Chair_wgm1 · 2023-12-06

**Metareview:**

*Summary*: This paper aims to mitigate the high-frequency oscillation issues in reinforcement learning, especially when applied to real. This paper first benchmarks existing methods in two categories, loss regularization, and architectural methods, in various simulation and real-world environments. The benchmark study finds that the proposed hybrid methods, particularly LipsNet + CAPS or L2C2, perform well in those environments in terms of policy performance and smoothness.

*Strength*: (1) High-frequency oscillation is indeed a common phenomenon in RL. (2) Thorough benchmarking in various environments with quantitative analysis.

*Weakness*: (1) The motivation is unclear and indirect. Making policy smoother is never the goal whereas improving performance is. Many highly oscillative classic controllers perform well in real, such as bang-bang control and MPC using L1 cost. This paper needs to further explain why and how oscillation is the key issue. In particular, considering sim-2-real, actuator delay, and bandwidth limit might play a much more important role. (2) The algorithmic contribution is little and relatively incremental. (3) The proposed hybrid methods do not consistently perform better in real scenarios, especially in terms of the cumulative return. Even for smoothness, the advantages are dominated by large variance.

**Justification For Why Not Higher Score:**

See the weakness part.

**Justification For Why Not Lower Score:**

See the strength part.

---

### Decision · Program_Chairs · 2024-01-16

Reject